# Prior exposure to antiretroviral therapy among adult patients presenting for HIV treatment initiation or reinitiation in sub-Saharan Africa: a systematic review

Mariet Benade,[1,2] Mhairi Maskew,[2] Allison Juntunen,[1] David B Flynn,[3] Sydney Rosen  [1,2]

¹Department of Global Health, Boston University School of Public Health, Boston, Massachusetts, USA
²Health Economics and Epidemiology Research Office, University of the Witwatersrand Faculty of Health Sciences, Johannesburg, Gauteng, South Africa
³Alumni Medical Library, Boston University School of Medicine, Boston, Massachusetts, USA

**Correspondence to**
Dr Sydney Rosen;
sbrosen@bu.edu

## ABSTRACT

**Objectives** As countries have scaled up access to antiretroviral therapy (ART) for HIV, attrition rates of up to 30% annually have created a large pool of individuals who initiate treatment with prior ART experience. Little is known about the proportion of non-naïve reinitiators within the population presenting for treatment initiation.

**Design** Systematic review of published articles and abstracts reporting proportions of non-naïve adult patients initiating ART in sub-Saharan Africa.

**Data sources** PubMed, Embase Elsevier, Web of Science Core Collection, International AIDS Society conferences, Conference on Retroviruses and Opportunistic Infections conferences.

**Eligibility criteria** Clinical trials and observational studies; reporting on adults in sub-Saharan Africa who initiated lifelong ART; published in English between 1 January 2018 and 11 July 2023 and with data collected after January 2016. Initiator self-report, laboratory discernment of antiretroviral metabolites, and viral suppression at initiation or in the medical record were accepted as evidence of prior exposure.

**Data extraction and synthesis** We captured study and sample characteristics, proportions with previous ART exposure and the indicator of previous exposure reported. We report results of each eligible study, estimate the risk of bias and identify gaps in the literature.

**Results** Of 2740 articles, 11 articles describing 12 cohorts contained sufficient information for the review. Proportions of initiators with evidence of prior ART use ranged from 5% (self-report only) to 53% (presence of ART metabolites in hair or blood sample). The vast majority of screened studies did not report naïve/non-naïve status. Metrics used to determine and report non-naïve proportions were inconsistent and difficult to interpret.

**Conclusions** The proportion of patients initiating HIV treatment who are truly ART naïve is not well documented. It is likely that 20%–50% of ART patients who present for ART are reinitiators. Standard reporting metrics and diligence in reporting are needed, as is research to understand the reluctance of patients to report prior ART exposure.

**PROSPERO registration number** CRD42022324136.

## STRENGTHS AND LIMITATIONS OF THIS STUDY

⇒ Comprehensive review using multiple databases and conference abstract archives.
⇒ Focused on recent data in the era since access to antiretroviral therapy for HIV became universal, increasing the potential for reinitiation of treatment.
⇒ Used best practices to develop search string, with multiple reiterations based on initial findings.
⇒ Recognised a wide range of reporting indicators for prior exposure to antiretrovirals medications.
⇒ Limited by the lack of standardised reporting of prior antiretroviral exposure.

## INTRODUCTION

The successful scale-up of access to antiretroviral therapy (ART) for HIV treatment in sub-Saharan Africa has produced a growing population of patients who have interrupted or stopped treatment sometime since they started, either permanently or temporarily. While very recent numbers on attrition from ART programmes are scarce, retention in care rates for the region were reported to average 78% at 12 months after treatment initiation in a review published in 2015, suggesting that for a cohort of patients initiating in any given year, nearly a quarter have been lost from care 1 year later.[1] Many of these lost patients, however, proceed to 'reinitiate' treatment in the months or years after dropping out of treatment programmes.[2] Two estimates posit the extent of treatment reinitiation. The first, from the US President's Emergency Plan for AIDS Relief (PEPFAR), reported that more than 580 000 patients returned to care after a treatment interruption in just the quarter from July to September 2020 in the countries that PEPFAR supports.[3] The second was from the Western Cape Province of South Africa,

where among the subset of patients whose CD4 counts were less than 50 cells/mm³, the proportion presenting for initiation with prior treatment experience rose from 14% to 57% between 2008 and 2017.[4]

Outside of indirect estimates such as those mentioned above, little is known about the actual proportions of non-naïve patients among all those presenting for ART initiation. Accurate data are difficult to obtain, largely because most HIV medical record systems neither distinguish between naïve and reinitiators nor allow tracking from one healthcare facility to another or over long intervals of inactivity. In most countries, a patient who originally initiated ART at one facility and then dropped out of care can easily present as a new patient at a nearby facility and be assumed to be ART naïve. Self-reported information about a patient's naïve or non-naïve status may be unreliable, because patients who are known to have stopped treatment may be reprimanded, provided poorer service by healthcare facility staff or required to participate in multiple adherence training sessions, creating an incentive to present oneself as a new patient regardless of prior experience.[5]

As the number of reinitiators continues to increase, understanding the proportion and characteristics of ART initiators who are not treatment-naïve is an important step in improving overall HIV treatment outcomes. By definition, most treatment reinitiators previously faced barriers to retention in care that they were unable to overcome in time or sufficiently to sustain continuity of treatment. Common barriers to retention include logistical challenges, such as transport costs, psychosocial deterrents, such as stigma, and personal preferences,[6 7] and these barriers may become more prohibitive for patients who have already withdrawn from care once. Achievement of long-term retention in care targets may thus require that healthcare systems differentiate interventions and services for reinitiators from those offered to naïve initiators. Reinitiators also comprise an increasing proportion of patients presenting with advanced HIV disease,[4] who require additional care beyond simple ART initiation.

As national treatment programmes mature, the proportion of ART initiators who are non-naïve will continue to grow, making the few available earlier estimates obsolete. To help fill the gap in empirical evidence on current proportions of naïve and non-naïve ART initiators, we conducted a systematic review of recently published or presented (2018 and later) peer-reviewed reports in sub-Saharan Africa that directly or indirectly presented data on reinitiation rates.

## METHODS

Guided by the Cochrane Handbook for Systematic Reviews of Interventions,[8] we conducted a systematic review of peer-reviewed publications and published conference abstracts that reported on prior exposure to ART among adult patients presenting for HIV treatment in sub-Saharan Africa. The review was registered on the International Prospective Register of Systematic Reviews (PROSPERO; CRD42022324136) (online supplemental file 1). We report our findings in accordance to the Preferred Reporting Items for Systematic Reviews and Meta-Analyses (PRISMA) 2020 guidelines.[9]

### Search strategy, study selection and data extraction

For this review, our primary outcome was the proportion of adults presenting for ART initiation (initially or after interruption) in public sector HIV treatment programmes in sub-Saharan Africa who are not ART naïve. To indicate prior ART use, we accepted self-reported questionnaire responses, rates of viral load suppression at ART initiation (suggesting previous ART use), medical record evidence (eg, a prior viral load test) and biological measurements of the presence of antiretroviral (ARV) metabolites in blood, hair or urine specimens at or prior to treatment initiation. Inclusion and exclusion criteria for the review are shown in online supplemental file 4.

To identify potential sources, we developed a search string with the assistance of a medical librarian. The search syntax included variations of the terms HIV, treatment, ART, retention and adherence, limited to sub-Saharan Africa. During the course of the review, we made three revisions to the original protocol as submitted to PROSPERO, in order to maximise the potential for finding relevant sources. Specifically, we (1) included studies focusing on pregnant women starting ART in prevention of mother-to-child transmission (PMTCT) programmes; (2) added 'undisclosed' and 'retention' as search terms, as shown in online supplemental tables S2 and S3; and (3) included studies that required a minimum duration of ART for enrolment, as explained below. Search strings can be found in online supplemental table S2.

We searched the PubMed, Embase (via Elsevier) and Web of Science Core Collection databases and published abstracts for the International AIDS Conference, International AIDS Society Conference on HIV Science and Conference on Retroviruses and Opportunistic Infections with a search string developed to identify English-language publications which reported on HIV treatment initiation in sub-Saharan Africa from 1 January 2018 to 11 July 2023. We further limited our search to articles from which the majority of the data were generated in 2016 or later, as this was when universal treatment access became common in the region and previous use became more likely. To capture the era of universal treatment access and allow time for guideline adoption, we considered studies published after 1 January 2018. Final searches conducted on 11 July 2023 updated our search phrases to include country names of countries in sub-Saharan Africa, using a prebuilt search strategy created by the Canadian Health Libraries Association.[10 11] Searching for abstracts using Google Advanced search and limiting results to each conference's domain did not yield results, as such we scanned the abstract booklets for abstracts that reported on populations that were initiating ART treatment and

searched conference archives with the keywords 'initiation', 'naïve', 'reinitiate', 'newly' and 'experience'.

We included cohort, cross-sectional, case–control and interventional studies that reported primary data on initiation of ART in the adult (≥18 years old) population, including for PMTCT. We included indexed preprints but excluded unpublished reports and publications in languages other than English. We also excluded commentaries, modelling studies and other sources that did not report primary data. While we excluded systematic reviews and meta-analyses, we manually searched existing systematic reviews for additional, non-duplicate references to be included. Where more than one publication reported on the same patient cohort, we chose the one that was either most recent or provided the most relevant data.

Because we were interested specifically in reports of the proportion of patients presenting for ART initiation or reinitiation in routine care who were naïve and non-naïve, we excluded studies that stated that prior ART experience was an exclusion criterion for the study, with two exceptions. If a study included only participants who self-reported as naïve but were then found to have evidence of previous ARV exposure, we included that proportion as a result, accepting that it explicitly omits patients who excluded themselves because of prior exposure and thus almost certainly reflects an underestimate of the true population prevalence of previous exposure. We also included studies that only enrolled patients who had achieved a specified duration of follow-up on ART—whether 1 month or several years—despite the fact that these studies would have missed patients who were lost from care prior to reaching that specified duration.

All peer-reviewed references identified using the respective search strings from PubMed, Embase and Web of Science were imported into Rayyan QCRI, where deduplication occurred. An initial, independent, blinded review (reviewers were not aware of each other's decisions) of the titles and abstracts was conducted by three study team members (MB, AJ and SR) using Rayyan QCRI. A full-text review was then conducted for all publications remaining after the initial review by two study team members (MB, AJ or MB, SR), with conflicts resolved through discussion. Reasons for excluding publications were recorded during the full-text review. As a quality check, one author (SR) also checked a sample (10%) of the excluded sources against exclusion criteria. At each stage of the review process, any conflicts between reviewers were assessed and resolved though consensus of three authors (MB, AJ and SR). The results of the search were documented in accordance with the PRISMA-S reporting checklist (online supplemental file 2).

We created a data extraction tool to capture study and sample characteristics, proportions with previous ART exposure and the type of indicator of previous exposure reported (eg, self-report, laboratory results).

## Outcomes and analysis

Our outcome of interest was the proportion of ART initiates who were treatment-naïve at ART initiation, defined as a patient presenting for initiation of ART who has never previously taken ART for treatment of HIV ('new initiator'), compared with the proportion who were treatment experienced at ART initiation, defined as a patient presenting for initiation of ART who had previously taken ART for HIV treatment but had interrupted that therapy for a minimum of 3 months ('reinitiator'). We accepted each paper's source of information about participants' status: self-report, medical record review, viral suppression or laboratory tests for ARV metabolites and report that source in our results.

To evaluate the data, we first report each paper's outcome, with descriptive information regarding the population and setting to which the results apply. As described in the PROSPERO registration record, we had intended to estimate pooled results for individual countries and populations and to stratify by patient and facility characteristics, but we ultimately identified too few eligible sources to allow for any pooled or stratified analysis.

Quality of the eligible studies was assessed using the Joanna Briggs Institute Critical Appraisal Checklists.[12]

## Patient and public involvement

None.

# RESULTS
## Sources identified

The results of the systematic search are shown in figure 1. A total of 2740 non-duplicate abstracts of peer-reviewed journal articles and 9 abstracts from the selected conferences were screened. Systematic reviews that were screened are listed in online supplemental file 3. After the initial title and abstract review, 2361 articles and abstracts were excluded, leaving 379 documents for full-text review. During the full-text review, an additional 368 documents were excluded. Reasons for exclusions are reported in online supplemental table S3. The primary reason for exclusion was lack of information on the naïve or non-naïve status of participants.

In total, 11 peer-reviewed articles were retained in the final dataset for the full review, including 1 that reported data from 2 countries and will be included in our analysis as 2 studies, creating a total of 12 sets of results. The studies are described in table 1.

Six of the studies in table 1 were conducted in South Africa and one per country in Botswana, the Democratic Republic of Congo, Ethiopia, Kenya and Zambia. Nine of the twelve reported baseline data from a clinical trial conducted for other purposes, while three were observational studies. All were very or somewhat small in size, enrolled adult patients presenting or having previously presented for routine ART initiation at public sector clinics, and, with the exception of Pry et al[13], collected

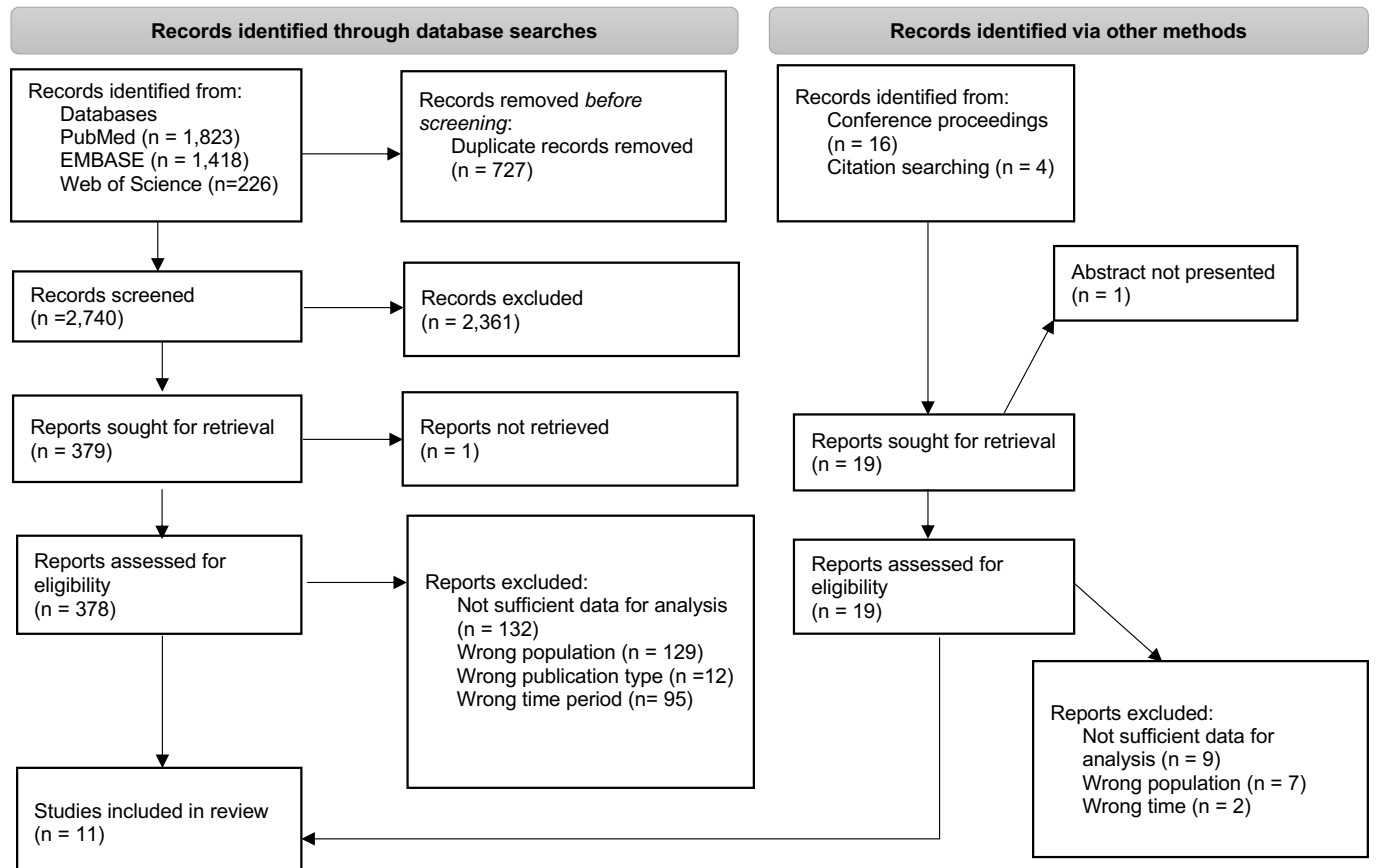

**Figure 1** Preferred Reporting Items for Systematic Reviews and Meta-Analyses flowchart.

most or all data prior to the disruption in service delivery caused by COVID-19 in early 2020.[14] Specific populations enrolled varied by study, though only one, Kunzweiler et al[15], was limited to a non-general adult population.

### Proportions of non-naïve patients

In table 2, we report the proportion of patients in each study reported to be non-naïve when presenting for ART initiation.

The proportion of patients presenting for ART initiation who were reported to be non-naïve ranged from 2% to 53%. It is important to note that the results shown in table 2 are not strictly comparable to one another, however, due to the different data sources, populations and relevant exclusions listed in the right-hand column of the table. In Maskew (2020)[16] and Rosen (2019)[17], non-naïve patients were allowed to enrol as long as they had been off ART for at least 3 months. Nevertheless, these studies, which are the only ones that relied entirely on participant self-report, reported the lowest proportion of non-naïve participants. Barnabas (2022)[18], which used both self-report and the facility's medical records, reported a much higher proportion with prior ART experience (50%) in the study's very small sample of initiators; rates varied between 33% and 66% previous exposure in the control and intervention arms, but that difference comprised only one individual in absolute terms. Two other studies in table 2, Mavhandu-Ramarumo (2021)[19]

and Sithole (2021)[20], explicitly excluded self-reported non-naïve patients; all patients enrolled in these studies claimed not to have been on ART previously. The relatively high proportions of non-naïve patients in these studies are thus still minimum estimates of the true proportion of ART initiators with prior treatment exposure at the study clinics, as anyone who admitted to prior use of ARVs will have been excluded from study enrolment. Similarly, Kunzweiler et al[15] and Dorward et al[21] excluded patients who were lost to follow-up before 6 months after ART initiation, and thus may have underestimated the proportion reinitiating if non-naïve patients are more likely than naïve patients to drop out of care. Study participants were also drawn from diverse populations of individuals with HIV, including those identified in a community campaign (Lebelonyane 2020)[12], in a household survey as part of a randomised trial (Sithole 2021[20]), or as routine, walk-in presenters at clinics (Buju 2022[22], Maskew 2020[16], and Rosen 2019[17]).

Because all the eligible studies were small and/or had primary outcomes other than the proportion of naïve, little stratification of results by facility or patient characteristics was included. Mavhandu-Ramarumo (2019)[19] reported that 7 of the 8 (88%) males in the sample population had evidence of prior ART exposure, compared with only 49% of the 34 females. In contrast, Sithole (2021)[20] found that women (37%) were more likely to

**Table 1** Studies included in the review

| Source (alphabetical by country) | Country and location | Study design and source of data | Study population | Dates of presentation for ART initiation | Sample size | Sex (% female) | Age (median, IQR) |
|---|---|---|---|---|---|---|---|
| Lebelonyane et al (2020)[12] | Botswana (national sample) | Subset of baseline data for intervention arm of cluster randomised trial of HIV prevention | Adults found to be HIV positive during community HIV prevention campaign and linked to HIV care | June 2016–March 2018 | 800 | 55% | 33 (26–41) |
| Buju et al[22] | Democratic Republic of Congo (Bunia) | Observational prospective cohort of patients receiving dolutegravir | Adults presenting for ART initiation or ongoing ART treatment at all ART clinics in Busia | July 2019–July 2021 for those initiating during the study; earlier for those already on ART | 177 | 69% | 39 (12)* |
| Genet et al[23] | Ethiopia (Northwest Ethiopia) | Observational, cross-sectional study of first-line treatment failure | Adults† who had received at least 6 months of ART | May–October 2017 | 430 | 58% | 38 (12–67)‡ |
| Kunzweiler et al[15] | Kenya (Kisumu) | Observational prospective cohort | Adult men who have sex with men | August 2015–September 2016 | 63 | 0% | 27 (22–32) |
| Rosen et al[17] | Kenya (Kericho, Kapsabet and Kombewa counties) | Baseline data for intervention arm of clinical trial of same-day ART initiation | Adults presenting for ART initiation at three public sector hospitals | July 2017–April 2018 | 477 | 58% | 36 (29–44) |
| Barnabas et al[18] | South Africa (KwaZulu Natal Province) | Baseline data from clinical trial of fee for ART home delivery | Adults living in study community and presenting for ART initiation | October 2019–January 2020 | 6 | 46%§ | 36 (31–43)§ |
| Dorward et al[21] | South Africa (KwaZulu Natal Province) | Baseline data for randomised controlled trial for point-of-care HIV viral load testing | Adults clinically stable on ART and due for 6-month viral load testing | August 2016–February 2017 (est) | 390 | 60% | 32 (27–38) |
| Maskew et al[16] | South Africa (Gauteng Province) | Baseline data for intervention arm in clinical trial of same-day ART initiation | Adults presenting for ART initiation at three public sector clinics | March–September 2018 | 296 | 64% | 35 (30–44) |
| Mavhandu-Ramarumo (2019)[19] | South Africa (Limpopo Province) | Baseline samples from clinical trial of drug resistance | Adults presenting for ART initiation at three public sector clinics | 2017–2019 | 77 | 90% | 35 (27–42) |
| Rosen et al[17] | South Africa (Gauteng Province) | Baseline data for intervention arm of clinical trial of same-day ART initiation | Adults presenting for ART initiation at three public sector clinics | March–July 2017 | 600 | 63% | 34 (29–41) |
| Sithole et al[20] | South Africa (KwaZulu Natal Province) | Subset of baseline data from clinical trial of home-based ART initiation | Non-pregnant adults presenting for ART initiation at two public sector clinics, with CD4>100 and no active TB | February 2018–November 2018 | 193 | 60% | Not reported |
| Pry et al[13] | Zambia (Lusaka) | Subset from implementation trial that used viral load testing at baseline | Adults diagnosed and initiating ART at two government facilities | May 2021–March 2022 | 248 | 63% | 30 (25–37) |

*Mean (SD).
†Study included children under 18, but they comprised<7% of the study sample.
‡Mean (minimum–maximum).
§Study enrolled a larger sample of patients already on ART—only six were reported as initiating or reinitiating. Characteristics reported are for the entire study population, not solely for a subsample of six who initiated treatment.
ART, antiretroviral therapy.

**Table 2** Proportions of cohorts reported to be non-naïve at ART initiation

| Source | Source of data | Proportion non-naïve at initiation | Inclusion criteria related to prior ART exposure | Comments |
|---|---|---|---|---|
| Lebelonyane et al (2020)* | Not stated; only indicated as participants who had 'previous ART treatment'. | 54/800 (7%) | Sample included 16–17 year olds; minors may be less likely than adults to have had an opportunity for prior ART exposure.† | Participants were identified through community-based testing and referred for ART; sample does not represent routine walk-in ART initiation population. |
| Buju et al[22] | Viral load suppressed at ART initiation. | 93/177 (52%) | None. | Enrolled both initiators and patients already on ART; results presented are only for initiators. |
| Genet et al[23] | Self-report | 90/430 (21%) | Sample included 12–17 year olds‡; minors may be less likely than adults to have had an opportunity for prior ART exposure. Participants required to have completed 6 months on ART; patients lost to follow-up before reaching 6 months were excluded. | Enrolled patients younger than 18. Those on second line ART were excluded; patients eligible for second line treatment may be more likely to be reinitiators. |
| Kunzweiler et al[15] | Viral load suppressed at ART initiation. | 19/63 (30%) (13 of 19 with suppressed viral load at ART initiation self-reported being ART naïve.) | Excluded nine participants who were lost to follow-up before reaching 6 months on ART. | Also excluded three patients who did not have viral load test results. |
| Rosen et al[17] | Self-report | 18/240 (8%) | Included self-reported reinitiators who had interrupted ART for ≥3 months. | Patients who self-reported that they had previously initiated ART but had stopped for ≥3 months were eligible; those who self-reported that they had interrupted for <3 months were excluded. |
| Barnabas et al[18] | Self-report and no existing record of ART at the study clinic. | 3/6 (50%) | Participants required to consent to a trial of payment for home ART; only clinically stable patients enrolled. | Enrolled both initiators and patients already on ART; results presented are only for initiators. |
| Dorward et al[21] | Self-report | 18/390 (5%) | Participants required to have completed 6 months on ART; patients lost to follow-up before reaching 6 months were excluded. | |
| Maskew et al[16] | Self-report | 33/296 (11%) | Included self-reported reinitiators who had interrupted ART for ≥3 months. | Patients who self-reported that they had previously initiated ART but had stopped for ≥3 months were eligible; those who self-reported that they had interrupted for <3 months were excluded. |
| Mavhandu-Ramarumo (2019) | Laboratory assay of blood and/or hair sample for presence of TDF, EFV and/or FTC metabolites. | 41/77 (53%) | All participants self-reported as ART naïve. | Stated enrolment criteria included naivete; those who had been on ART previously may have self-screened out of this study. |
| Rosen et al[17] | Self-report | 7/298 (2%) | Included self-reported reinitiators who had interrupted ART for ≥3 months. | Patients who self-reported that they had previously initiated ART but had stopped for ≥3 months were eligible; those who self-reported that they had interrupted for <3 months were excluded. |

Continued

**Table 2** Continued

| Source | Source of data | Proportion non-naïve at initiation | Inclusion criteria related to prior ART exposure | Comments |
|---|---|---|---|---|
| Sithole et al[20] § | Undetectable viral load at ART initiation. | 62/193 (32%) (other outcomes reported: 42/193 (22%) had medical record evidence of prior ART use; 37/193 (19%) had detectable antiretroviral metabolites in blood.) | All participants self-reported as ART naïve. | Stated enrolment criteria included naivete; those who had been on ART previously may have self-screened out of this study. Participants were identified through community-based testing; sample does not represent routine walk-in ART initiation population. Primary DO-ART trial excluded 588 participants out of 2479 who were virally suppressed at time of initiation and 66 who were found to already be on ART[31] |
| Pry et al[13] | Undetectable viral load at ART initiation. | 66/248 (27%) (other outcomes reported: among 57 participants with suppressed viral load who completed survey on prior ART use, 14% reported previous ART.) | None. | Survey on silent transfers (self-reported prior ART use) was conducted at a follow-up visit after it was determined that participants had a suppressed viral load at initiation. |

*A separate publication by the same author team, reporting data that overlapped with those reported in this study, stated that "22% of advanced HIV patients had previously been on ART".[32]
†16–17 year olds receive treatment as adults in this setting.
‡Minors under age 18 comprised <7% of the study sample.
§The parent study on which this study was based.
ART, antiretroviral therapy; EFV, efavirenz; FTC, emtricitabine; TDF, tenofovir.

have evidence of undisclosed ART use than men (25%). Other characteristics that were associated with undisclosed ART use by undetectable viral load were younger age (35% in 18–29 year olds, 30% in 30–49 year olds and 23% among those >50 years) and living with a partner who was HIV positive (44% compared with 37%; adj OR 1.94 (95% CI 0.95 to 3.96)). Kunzweiler (2018)[15] results apply specifically to men who have sex with men. Pry 2023[13] found that women ≥40 years had the highest probability of being non-naïve (42%, 95% CI: 39.3% to 44.3%), while men aged 18–24 years had a baseline probability of just 12% (95% CI: 4.6% to 19.0%). Age was associated with higher rates of viral suppression, even when adjusting for sex, marital status, education or facility of initiation. Being married and being female were both also associated with significantly increased adjusted prevalence ratios. Other studies did not provide information for stratification by facility or patient characteristics.

## Quality of evidence

Each of the studies included in the review either presented baseline enrolment data from a randomised controlled trial or observational data. Outcomes after ART initiation were not relevant for our review, which looked only at the status of patients at baseline (ART initiation). For Maskew et al[16] and Rosen et al[17], data on self-reported naïve status were limited to the intervention arm. The

setting, inclusion criteria and baseline characteristics for these studies were clearly defined. Mavhandu-Ramarumo (2019)[19], Lebelonyane (2020)[12], Dorward et al[21], Barnabas et al[18] and Sithole et al[20] also sufficiently defined inclusion criteria, study setting and baseline characteristics.

We assessed Kunzweiler et al[15], Genet (2021) et al[23], Buju(2022)[22] and Pry et al[13] as observational studies. Kunzweiler et al[15] used snowball sampling, as is frequently done among key populations at high risk of stigma. All four papers described the study setting and sample population of interest clearly, including age, sex and clinical characteristics of HIV presentation.

As is indicated in table 2 and discussed further below, each of the studies included in this review used a different indicator of prior ART exposure (non-naivete), and most had limitations as to their accuracy and/or the representativeness of their populations.

## DISCUSSION

We systematically reviewed peer-reviewed evidence on the proportions of patients presenting for treatment initiation in sub-Saharan Africa who are or are not ART naïve. The proportions non-naïve in the sources we found ranged from a low of 2% using self-report only to a high of 53% based on a laboratory analysis of ARV metabolites

in blood and hair samples of patients who self-reported to be naïve.

Perhaps the most striking finding of this review is the sheer lack of published evidence to answer our research question. Despite a comprehensive search of the literature published between 2018 and 2022, and including data since 2016, we identified only 11 sources and 12 cohorts that reported this information, and most included it in only in passing. Half the studies were conducted in South Africa and were relatively small in size; most of those from other countries provided very little detail. Based on the published and presented research alone, it is fair to say that very little is known about the true proportion of ART initiators who are not treatment-naïve in South Africa, and almost nothing is known about the rest of the region or about specific subpopulations or risk groups. While it is possible that more information is available to programme managers who have access to routinely collected medical record data, nothing in the literature suggests that such information is being generated on a large scale or, more important, used for programme improvement.

During our search, we made a concerted effort to find additional eligible sources, in the hope that there would be more data to review and analyse. This included adding additional search terms, including data on pregnant and postpartum women, and reviewing reference lists from relevant systematic reviews. We reviewed an unusually large number of full-text manuscripts in the hope that they would include proportions of naïve and non-naïve in their cohort descriptions (typically table 1) even though there was no indication of this in the abstract.

Unfortunately, most of the sources that originally appeared promising were found to be ineligible, for various reasons. Most simply did not report on baseline naïve/non-naïve status at ART initiation, using any indicator. Some intentionally excluded non-naïve patients prior to enrolment and then reported all participants as naïve without further investigation. For studies that explicitly screened out non-naïve participants, we considered calculating proportions non-naïve based on reported numbers of potential participants included and excluded, but we realised that many studies only applied the non-naïve criterion after screening potential participants out for other reasons. We thus could not safely rely on numbers screened out due to non-naïve status for our numerator and had to exclude those papers. Even some of the papers we did include either provided only a passing reference about prior ART exposure, making us uncertain that we interpreted them correctly (eg, Lebelonyane *et al*, 2020), or explicitly excluded patients lost to follow-up in the early treatment period (eg, Kunzweiler *et al*[15] and Dorward *et al*[21]). Finally, several studies were excluded because we could not interpret the data reported or had doubts about the accuracy of prior exposure data based on the data sources used, even beyond the limitations discussed below.

The small number of eligible papers we did find offers some useful information. They used several different indicators for identifying non-naïve patients, and the indicators produced results that are consistent with their expected accuracy. Maskew *et al*[16] and Rosen *et al*[17], which relied solely on self-report, and Lebelonyane *et al* (2020), for with the source of data is unknown, found the lowest proportions non-naïve. Sithole *et al*[20], which excluded a priori anyone self-reporting prior utilisation, reported that 32% of patients had evidence of prior use based on being virally suppressed. Buju *et al*[22] and Kunzweiler *et al*[15] reported that 52% and 30%, respectively, of patients presenting for initiation already had suppressed viral loads, suggesting prior ARV use. Mavhandu-Ramarumo (2019), which also excluded a priori anyone admitting prior utilisation, using the most rigorous methodology with both blood and hair samples, estimated 53% of patients had prior ART exposure. The large observed difference between males and females and the small number of males in this study, however, suggest caution in applying the results to male patients. The expectation of viral rebound within 4 weeks of treatment interruption[24] suggests that viral suppression underestimates the true proportion non-naïve, as suppression as an indicator only captures recent treatment interrupters. Similarly, metabolite tests capture a maximum of about 90 days' prior exposure; anyone who interrupted treatment more than 3 months prior to study enrolment would be missed in the count of non-naïve initiators.[25]

Based on the results of the studies that used laboratory tests, we assume that those that relied on self-report alone—eg, Maskew *et al*[16], Rosen *et al*[17] and Dorward *et al*[21]—underestimated the true proportion of participants who were non-naïve at initiation. Barnabas (2023), which combined self-report and same facility record review, is an outlier among the studies using self-report, but the very small sample size of initiators (n=6) suggests caution in drawing conclusions from it. Although study inclusion criteria may have biased the samples in Sithole *et al*[20] and Mavhandu-Ramarumo (2019), these studies, together with Buju *et al*[22], Kunzweiler *et al*[15], Genet *et al*[23] and Pry *et al*[13], suggest that it is reasonable (and conservative) to conclude that between 20% and 50% of ART patients—and likely at least 30%—who present for ART are reinitiators. This proportion can be expected to increase with each passing year, as the number of truly naïve HIV-positive individuals declines. If this is so, then reinitiators comprise an important subpopulation whose needs are likely to differ from those of naïve initiators and to whom service delivery should be tailored. This is especially important as recent evidence shows that mortality is significantly higher among PLHIV who interrupt and then reinitiate treatment, especially if the interruption occurred within 6 months of ART initiation.[26]

As is evident from the discussion above, this review had several limitations. First, while we believe that our search of the peer-reviewed, published literature and abstracts was thorough, the lack of standard terminology for describing prior ART exposure hampered the creation of precise search strings, and it is possible that some sources

were missed. Second, we found information from only 6 of sub-Saharan Africa's 46 countries, and 6 of our 12 observations were from a single country, South Africa. Since each country in sub-Saharan Africa has a different experience with attrition from ART and approach to reinitiation, results may not be generalisable. Third, as explained in the introduction, even such data as are available tend to be incomplete, due to the limitations of self-reporting and of existing medical record systems.

Fourth, the wide range of results identified may reflect study methodologies, but it may also indicate substantial geographic diversity in outcomes that we cannot address with the data available. We speculate that additional relevant data are collected by programme managers, ministries of health and others but are either not analysed to answer our research question or simply not published and therefore not accessible. Fifth, the small number of eligible sources, small sample sizes and heterogeneity of research methods made it impossible to aggregate the results or produce meaningful summary statistics, beyond the range discussed above. Sixth, recall bias may be present in the studies that asked participants after 6 months or more on ART to self-report their naïve/non-naïve status at the time they initiated treatment. Seventh, none of the papers included reported the duration of the interval between a participant's prior ART experience and reinitiation. Some apparent reinitiators may in fact reflect unrecorded ('silent') transfers from one facility to another, without a medication interruption in between. This phenomenon may help explain the high proportion of 'initiating' patients who already have viral suppression in Buju et al[22], for example.

Finally, the fact that 9 of 12 cohorts included were from clinical trials that provided patient compensation may have biased enrolments, though we cannot know how this may have affected patient enrolment or self-report of prior ART usage and the subsequent direction of the bias.

In addition to the sheer dearth of information available to answer our question, the search reported here revealed three important research priorities. First, there is a need for a standard terminology to describe patients with prior ART exposure and prior ART initiation experience. Even the binary terms 'naïve' and 'non-naïve' can be unclear if patients have previously used ARV medications for pre-exposure prophylaxis (PrEP) or prevention of vertical transmission. Terms such as 'ART experienced' and 'ART exposed' are often substituted for non-naïve, without specification of what they refer to. Prior ART use may be 'disclosed' or 'self-reported'. Similarly, the duration of treatment interruption that leads to 'reinitiation' is rarely specified, and 'reinitiation' and 're-engagement' are used interchangeably. We can assume that the patient returning to care after an interruption of less than 1 month is not likely to be regarded as a reinitiator, and a patient returning after an interruption of more than 1 year will likely be defined as a reinitiator. But what of patients with interruptions of 6 or 8 months? A common terminology for describing the phenomenon addressed

here would be of great assistance in understanding its magnitude.

Second, in view of the potentially very high proportion of reinitiators among 'new' ART patients, it is critical that researchers begin to report proportions of naïve and non-naïve as a standard variable when describing patient cohorts, even if data come solely from self-report. We cannot determine from the literature whether many studies do collect this information but omit it from their reports or if it has simply not been collected. We identified several papers that came close to indicating a proportion non-naïve but did not explain their findings clearly enough to include in this review, suggesting that many studies do indeed have access to the relevant information. The study mentioned in our introduction from South Africa's Western Cape Province, for example, provided detailed information about patients with advanced HIV disease and suggested that more than a third of all patients with very low CD4 cell counts had previously been on ART but were now off, but we could not calculate the overall proportion of non-naïve initiators from the data reported.[4] In earlier years, when the proportion of non-naïve patients was low because treatment programmes were still rapidly expanding, the question of prior ART experience may not have been a priority. In view of the results of the few studies available, it is clearly a priority now, for several reasons. First, retention in care will remain a challenge if those who disengaged from care previously remain at higher risk of disengaging from care again,[27] unless the obstacles to continuity in care have been identified and addressed. Second, if service delivery needs to differ between ART naïve and ART experienced clients, treatment outcomes may also be affected by the large number of non-naïve ART clients accessing services. Third, to the extent that resistance to first-line ARV medications remains a concern, non-naïve patients may face higher risks of poorer responses to these medications after reinitiation.

Finally, the phenomenon of large numbers of patients who decline to reveal prior ART use, even when asked directly, is concerning. For studies that intend to limit participation to naïve patients, stated exclusion of those who are non-naïve may encourage non-disclosure, to avoid being denied study enrolment on this basis. Even so, it appears likely that many patients opt to lie about prior exposure. Other research suggests that they have good reason for doing so, as clinics may refuse to reinitiate those who admit to prior default and/or may provide poorer service to them.[28–30] Creating a clinic and community atmosphere that promotes honesty about prior exposure should also be a priority.

In conclusion, while we recognise that simply knowing the proportion of non-naïve patients in a given population will not in itself improve the quality of service delivery, measuring the size of the problem is a critical step in creating momentum to development and implement interventions targeted specifically at reinitiators. Since these are patients who have already demonstrated

that they face obstacles to remaining in care, identifying and targeting them for appropriate services is a vital step in improving the outcomes of treatment programmes.

**Contributors** MM, SR and MB conceived of and designed the study. DBF, MB and SR created the search strategy. DBF conducted the searches. MB, SR and AJ identified and reviewed sources and extracted data. MB and SR analysed the data and drafted the manuscript. All authors reviewed and edited the manuscript. SR is the guarantor of this manuscript.

**Funding** Funding for the study was provided the Bill & Melinda Gates Foundation through INV-031690 to Boston University. The funder had no role in study design, data collection and analysis, decision to publish or preparation of the manuscript.

**Competing interests** None declared.

**Patient and public involvement** Patients and/or the public were not involved in the design, or conduct, or reporting, or dissemination plans of this research.

**Patient consent for publication** Not applicable.

**Provenance and peer review** Not commissioned; externally peer reviewed.

**Data availability statement** All data relevant to the study are included in the article or uploaded as online supplemental information.

**ORCID iD**
Sydney Rosen http://orcid.org/0000-0002-6560-2964

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
