## [Reviewer comments · BMJ Open]

ARTICLE DETAILS

TITLE (PROVISIONAL)	Prior exposure to antiretroviral therapy among adult patients presenting for HIV treatment initiation or re-initiation in sub-Saharan Africa: a systematic review
AUTHORS	Benade, Mariet; Maskew, Mhairi; Juntunen, Allison; Flynn, David; Rosen, Sydney

VERSION 1 – REVIEW

REVIEWER	Reynolds, John University of Miami Miller School of Medicine, Calder Memorial Library
REVIEW RETURNED	08-Mar-2023

GENERAL COMMENTS	This reviewers expertise is evidence synthesis database searching and methods, not infectious diseases or epidemiology, so I will limit my comments to the methods related to acquiring the evidence for the review. The title indicates that this is a systematic review, but the text describes it as a rapid review (lines 36, 118, 122). Please clarify the type of study this is. Rapid reviews should be undertaken when speed is important. Please explain the need to do a rapid review rather than a systematic review for this topic. According to the protocol, funding for this project was awarded in October 2021, and the manuscript submitted 14 months later, which should have given time for a full systematic review. The Strengths and Limitations box at line 65 describes the study as comprehensive. Rapid reviews are, by definition, less than comprehensive. The same contradiction is on line 118. Please clarify what kind of study was done. Reporting the full search strategies as an appendix and reporting results for each database in the PRISMA flow / Figure 1 is excellent. The search strategy seems inadequate, though. This is especially important since your conclusions focus on the lack of evidence for the topic. In addition, the PubMed search is not replicable. Running the search exactly as described, using the same dates, returns 1886 results, not 1396 as reported in the PRISMA flow chart. Reporting search strategies exactly as run is recommended. Which version of Embase was used? Ovid or Elsevier? Running your search as reported in Embase.com (Elsevier) returns only 216 results, while 854 are reported in Figure 1. Both versions of Embase use Emtree terms, which are not apparent in your strategy. Using those could have both increased
--

	the yield of relevant studies and reduced the number needed to screen. What databases were searched in Web of Science? Subscriptions to WoS vary by institution and there are many options available in the platform interface which can expand or focus results which were not reported as being used.. See the PRISMA-S checklist and explanatory documents for reporting the search strategy of an evidence synthesis review. http://www.prisma-statement.org/Extensions/Searching https://www.ncbi.nlm.nih.gov/pmc/articles/PMC7839230/ In line 138, the assistance of a medical librarian is mentioned. It is appropriate, and best practice, for librarians to conduct systematic review searches, and be acknowledged or listed as coauthor, depending on their level of involvement. The search strategy does not show signs that it was developed by a librarian with evidence synthesis expertise. It is a good search for a narrative review, but not for a systematic or rapid review. Considering the conclusions stating that there is insufficient evidence on the topic, this is particularly important. Using the appropriate subject headings and field codes for titles, abstracts, and keywords for each database will improve your results. Geographical searching for health studies is difficult and simply using terms for the entire region “Sub-Saharan Africa” will probably miss relevant reports. If a study was conducted in a single county, Ghana, for example, but “Sub-Saharan Africa” is not mentioned in the subject terms, keywords, title, or abstract, that study would have been missed. MeSH terms are automatically exploded in PubMed, so countries will be searched individually there, but since you only searched words in Embase and Web of Science, there is serious potential for missed evidence A comprehensive approach would be to follow the technique described here: Ayiku, L., Levay, P., & Hudson, T. (2021). The NICE OECD countries' geographic search filters: Part 1-methodology for developing the draft MEDLINE and Embase (Ovid) filters. Journal of the Medical Library Association, 109(2), 258-266. https://doi.org/10.5195/jmla.2021.978 and Ayiku, L., Hudson, T., Williams, C., Levay, P., & Jacob, C. (2021). The NICE OECD countries' geographic search filters: Part 2-validation of the MEDLINE and Embase (Ovid) filters. Journal of the Medical Library Association, 109(4), 583-589. https://doi.org/10.5195/jmla.2021.1224 That method is comprehensive, but time consuming and not appropriate for all circumstances. At the very least, including terms for all the applicable country names, major cities, and townships, as well as possibly provinces, states, and relevant health departments would be likely to improve the number of relevant results found. A less detailed strategy is sometimes warranted in a rapid review, but since the topic is geographical in nature, that should be reflected more accurately in the searching. The citation for rapid review guidelines mentioned in line 122, number 8, is described in the text as a WHO guideline, but is actually a protocol for PRISMA rapid review reporting guidelines. As a protocol, it offers no guidance, but is only a plan to develop new guidelines. (Dated 2014, it will be helpful to reviewers when it
--	---

	is completed). As a PRISMA document, when finished, it will only be guidance on reporting rapid reviews, not conducting them. There is WHO guidance on rapid review conduct here: Tricco, Andrea C., Langlois, Etienne. V., Straus, Sharon E., Alliance for Health Policy and Systems Research & World Health Organization. (2017). Rapid reviews to strengthen health policy and systems: a practical guide. World Health Organization. https://apps.who.int/iris/handle/10665/258698 Cochrane Rapid Reviews Methods Group also offers guidance to conduct rapid reviews, if there is justification for not doing a full systematic review. J Clin Epidemiol. 2020 Oct 14:S0895-4356(20)31146-X. doi: 10.1016/j.jclinepi.2020.10.007 . https://www.jclinepi.com/article/S0895-4356(20)31146-X/pdf PMID: 33068715 Due to imprecision of database cataloging and inconsistency in language used by authors, searching terms such as HIV, as opposed to HIV infections, and AIDS might yield useful results. Other synonyms such as HIV positive and others should be considered. Conversely, in line 1 of the PubMed search, the term "infections" by itself, will return off topic results for any kind of infection - ("HIV Infections"[Mesh] OR HIV Infection OR Infection) Line 36 states that only published reports were included. The Strengths and Limitations box at line 65 also refers to conference abstracts. Including conference abstracts is important, so these two statements should be reconciled. If you meant published conference abstracts and published articles, be specific. Searching conference abstracts is an excellent strategy for this topic. Please explain how this one done so that your results will be more transparent and replicable. The choice of study types to include (line 156) is good as is your use of citation searching in any systematic reviews found by your search. Those reviews could be cited or at least have the numbers indicated on the PRISMA flow chart/Fig.1 The screening methods described on 177 forward are solid and clearly explained. Line 184-5 states that the results of the search were documented in accordance with PRISMA-P. PRISMA-P is for reporting protocols. Did you use PRISMA-S? It recommends describing which version of databases you used you used (eg Embase) and which databases within multi-database platforms you used (eg Web of Science) which were not included. The search date is not mentioned. Although it could be the same as the end date for the searches, March 31, 2022, this is not clear. That date is nearly a year ago. Considering the small amount of evidence you found, rerunning the searches to find results published since March 2022 is recommended. Since you used Rayyan, which will identify duplicates, reviewing only new results would be simple.
--	---

	I recommend having the search strategy rewritten by a librarian experienced with evidence synthesis/systematic review searching. Using Rayyan to identify duplicates, you can avoid rescreeing. This should improve the recall of on-topic studies as well as finding any studies from the past year. You may end up with similar conclusions, but they will be based on better methods and evidence.
--	---

REVIEWER	U Van Zyl, Gert Stellenbosch University Faculty of Medicine and Health Sciences
REVIEW RETURNED	31-May-2023

GENERAL COMMENTS	The topic of prior antiretroviral exposure in patients presenting for ART initiation is very topical and important. Prior drug exposure may influence treatment outcomes and may result in a much higher virologic failure rate or rate of drug resistance than in patients who are truly therapy naïve. This is true even for high genetic barrier regimens: Whereas the incidence of virologic failure and especially drug resistance is exceeding rare in therapy naïve patients receiving first line regimens that include dolutegravir (DTG) as anchor drug, it is not uncommon in patients with virologic failure on a DTG containing regimen, who have had prior virologic failure on different regimens. Prior ART exposure could therefore result in lower real-life effectiveness of particular regimens than the expected efficacy observed in clinical trials of treatment naïve participants. The manuscript is overall well written. It followed a structured and clear search plan. The authors report heterogeneity in the methods to establish prior ART use. The proportion of prior ART use varied between 2% and 53%. In studies with lower proportions this may represent an underestimation due to relying on self-report. The authors also make an important comment, stating that reinitiating patients include patients who have experienced obstacles to care and may require services to improve their outcomes. Comment that could be considered for further improvement: Please add a brief comment on the potential impact of having a large proportion of non-naïve patients on apparent first-line ART treatment success.
---

REVIEWER	Dorward, Jienchi University of Oxford, Nuffield Department of Primary Care Health Sciences
REVIEW RETURNED	06-Jun-2023

GENERAL COMMENTS	Thank you for opportunity to review this systematic review of studies assessing the proportion of people who are ART naïve out of those initiating ART. This is a well written study of an important topic and appears to have been well conducted. I have few mainly minor comments MAJOR Throughout the manuscript, It would help to be a bit clearer about the different measures used in the studies, and the difference in what they are measuring ie currently or recently on ART (measured by suppressed VL and/or measured ART metabolites) versus those who at some point have been on ART (which could
---

	have been a long time in the past and so no ART metabolites would be detected). I realise that the lack of studies using the different methods means that analysis/stratification based on these different measures is not possible, but it would help with moving the field forwards if this difference was clearly set out in the introduction, and discussed in the discussion. MINOR Methods: Why were only studies after 2018 included? If Universal treatment started in 2016, then it is feasible that useful data could have been reported in 2017 Commendable effort with full text review of >300 manuscripts! I am also surprised that there are so few relevant studies. I'm not a search expert but would the search terms have captured studies that did not mention sub-Saharan Africa, but only mentioned a country by name? Line 107 "By definition, treatment re-initiators previously faced barriers to retention in care that they were unable to overcome." – what about people who relocate – its not necessarily a barrier that they have to overcome? For those with suppressed viral loads, could there have been Table 2: "Sample included 16-17 year olds" is it possible to give the proportion who were this age in the sample, which will help reader to determine how important this is. Likewise for other studies which included children Table S4: Studies excluded for the wrong population: "Population for main analysis excluded those with ART experience or required a minimum follow-up period after treatment initiation." But it seems like some studies which excluded people who weren't ART naïve, or required minimum follow-up e.g. Dorward were included. A bit more explanation about why some were excluded and some included would help.
--	---

VERSION 1 – AUTHOR RESPONSE

Reviewer 1

Prof. John Reynolds, University of Miami Miller School of Medicine

Comments to the Author:

This reviewers expertise is evidence synthesis database searching and methods, not infectious diseases or epidemiology, so I will limit my comments to the methods related to acquiring the evidence for the review.

We thank the Reviewer for these detailed recommendations and appreciate that our manuscript is stronger as a result.

Comment	Response
The title indicates that this is a systematic review, but the text describes it as a rapid review (lines 36, 118, 122). Please clarify the type of study this is. Rapid reviews should be undertaken when speed is important. Please explain the need to do a rapid review rather than a systematic review for this topic. According to the protocol, funding for this project was awarded in October 2021, and the manuscript submitted 14 months later, which should have given time for a full systematic review. The Strengths and Limitations box at line 65 describes the study as comprehensive. Rapid reviews are, by definition, less than comprehensive. The same contradiction is on line 118. Please clarify what kind of study was done.	Thank you for clarifying this. It is a systematic review. We have corrected this in the text.
Reporting the full search strategies as an appendix and reporting results for each database in the PRISMA flow / Figure 1 is excellent. The search strategy seems inadequate, though. This is especially important since your conclusions focus on the lack of evidence for the topic. In addition, the PubMed search is not replicable. Running the search exactly as described, using the same dates, returns 1886 results, not 1396 as reported in the PRISMA flow chart. Reporting search strategies exactly as run is recommended.	Thank you for highlighting this error. Our previous search strategy erroneously said that we included the term “infection”, which we had in fact removed due to its identifying articles about infections other than HIV. Removing that solves the discrepancy. We’ve updated our search strategy (incorporating other reviewer comments too) and have updated our reporting of our search strategy (Additional table S2) accordingly.
Which version of Embase was used? Ovid or Elsevier? Running your search as reported in Embase.com (Elsevier) returns only 216 results, while 854 are reported in Figure 1. Both versions of Embase use Emtree terms, which are not apparent in your strategy. Using those could have both increased the yield of relevant studies and reduced the number needed to screen. What databases were searched in Web of Science? Subscriptions to WoS vary by institution and there are many options available in the platform interface which can expand or focus results which were not reported as being used.. See the PRISMA-S checklist and explanatory documents for reporting the search strategy of an evidence synthesis review. http://www.prisma-statement.org/Extensions/Searching https://www.ncbi.nlm.nih.gov/pmc/articles/PMC7839230/	We used Embase Elsevier and Web of Science Core Collection. We have added the requested details about versions and databases. This section of the manuscript now reads, “...searched the PubMed, Embase (via Elsevier), and Web of Science Core Collection”. The Core Collection is comprised of the Science Citation Index Expanded (1965 to the present), the Social Sciences Citation Index (1965 to the present), the Arts & Humanities Citation Index (1975 to the present), the Conference Proceedings Citation Index (both versions, Science and Social Sciences & the Humanities from 1990 to the present), the Book Citation Index (both versions, Science and Social Sciences & the Humanities from 2005 to the present) and the Emerging Sources Citation Index (2018 to the present)

Comment	Response
	We have updated the search strategy to include the emtree terms. Upon re-conducting the search using country names as per the geographical hedge Canadian Health Libraries Association, confirming that “infection” was not added on its own, and updating it to the new search date, we identified 1,096 unique additional potential sources during the study time period. Of these, however, only two ,both published after the original search was completed, met all inclusion criteria. We have updated the results text and PRISMA diagram accordingly.
In line 138, the assistance of a medical librarian is mentioned. It is appropriate, and best practice, for librarians to conduct systematic review searches, and be acknowledged or listed as coauthor, depending on their level of involvement. The search strategy does not show signs that it was developed by a librarian with evidence synthesis expertise. It is a good search for a narrative review, but not for a systematic or rapid review. Considering the conclusions stating that there is insufficient evidence on the topic, this is particularly important. Using the appropriate subject headings and field codes for titles, abstracts, and keywords for each database will improve your results. Geographical searching for health studies is difficult and simply using terms for the entire region “Sub-Saharan Africa” will probably miss relevant reports. If a study was conducted in a single county, Ghana, for example, but “Sub-Saharan Africa” is not mentioned in the subject terms, keywords, title, or abstract, that study would have been missed. MeSH terms are automatically exploded in PubMed, so countries will be searched individually there, but since you only searched words in Embase and Web of Science, there is serious potential for missed evidence A comprehensive approach would be to follow the technique described here: Ayiku, L., Levay, P., & Hudson, T. (2021). The NICE OECD countries' geographic search filters: Part 1- methodology for developing the draft MEDLINE and Embase (Ovid) filters. Journal of the Medical Library Association, 109(2), 258-	We thank the reviewer for alerting us to this shortcoming in our search strategy. We have updated our search to include country names, using a hedge from Canadian Health Libraries Association that was developed by the InterTASC Information Specialists Group (specifically the ISSG Search Filter Resource), an association of health professionals who create assessments for organizations like NICE. Members of the InterTASC Information Specialists Group include Julie Glanville and Carol Lefebvre from Cochrane so this is a highly respected source. Geographical hedges for Systematic Reviews in the Health Sciences such as ours often include controlled vocabulary terms for the region (such as "Africa South of the Sahara"[Mesh]) combined with keyword searches for the individual countries. As sub-Saharan Africa has nearly 50 countries, each with multiple major cities and townships, states, provinces, and health departments, (for example, South Africa alone has 9 provinces, 52 districts, and 257 municipalities, each of which has its own department of health), it is not feasible to try to search for each one of these by name. We did, however, as mentioned above, add each country's name, and we thank the reviewer for this suggestion.

Comment	Response
266. https://doi.org/10.5195/jmla.2021.978 and Ayiku, L., Hudson, T., Williams, C., Levay, P., & Jacob, C. (2021). The NICE OECD countries' geographic search filters: Part 2-validation of the MEDLINE and Embase (Ovid) filters. Journal of the Medical Library Association, 109(4), 583-589. https://doi.org/10.5195/jmla.2021.1224 That method is comprehensive, but time consuming and not appropriate for all circumstances. At the very least, including terms for all the applicable country names, major cities, and townships, as well as possibly provinces, states, and relevant health departments would be likely to improve the number of relevant results found. A less detailed strategy is sometimes warranted in a rapid review, but since the topic is geographical in nature, that should be reflected more accurately in the searching.	
The citation for rapid review guidelines mentioned in line 122, number 8, is described in the text as a WHO guideline, but is actually a protocol for PRISMA rapid review reporting guidelines. As a protocol, it offers no guidance, but is only a plan to develop new guidelines. (Dated 2014, it will be helpful to reviewers when it is completed). As a PRISMA document, when finished, it will only be guidance on reporting rapid reviews, not conducting them. There is WHO guidance on rapid review conduct here: Tricco, Andrea C., Langlois, Etienne. V., Straus, Sharon E., Alliance for Health Policy and Systems Research & World Health Organization. (2017). Rapid reviews to strengthen health policy and systems: a practical guide. World Health Organization. https://apps.who.int/iris/handle/10665/258698 Cochrane Rapid Reviews Methods Group also offers guidance to conduct rapid reviews, if there is justification for not doing a full systematic review. J Clin Epidemiol. 2020 Oct 14:S0895-4356(20)31146-X. doi: 10.1016/j.jclinepi.2020.10.007 . https://www.jclinepi.com/article/S0895-4356(20)31146-X/pdf PMID: 33068715	As mentioned above, we agree with the Reviewer's comments that this should correctly be designated a systematic review, not a rapid review. We have updated the citation accordingly, using the PRISMA 2020 statement as our source.

Comment	Response
Due to imprecision of database cataloging and inconsistency in language used by authors, searching terms such as HIV, as opposed to HIV infections, and AIDS might yield useful results. Other synonyms such as HIV positive and others should be considered. Conversely, in line 1 of the PubMed search, the term “infections” by itself, will return off topic results for any kind of infection - ("HIV Infections"[Mesh] OR HIV Infection OR Infection)	Thank you, we've added information on the databases we searched to improve clarity. Given the population, which at this point consists of very few people living with advanced HIV disease meeting the clinical definition of AIDS at initiation of treatment, we did not include AIDS in our search terms. As per our updated, PROSPERO protocol published in October 2022, the term “infections” by itself was not included in our search strategy – the inclusion in the submitted search strategy table was an error on our part and we apologise for any confusion this might have caused.
Line 36 states that only published reports were included. The Strengths and Limitations box at line 65 also refers to conference abstracts. Including conference abstracts is important, so these two statements should be reconciled. If you meant published conference abstracts and published articles, be specific.	Thank you, we apologize for the imprecision. We did mean published conference abstracts and published articles and have corrected the text to indicate this.
Searching conference abstracts is an excellent strategy for this topic. Please explain how this one done so that your results will be more transparent and replicable.	Thank you, we initially used the search phrases we developed for our database searches to conduct a search through Google using advanced settings to limit it to the domain of each conference. This yielded poor results and as such we manually screened abstract booklets for sources that reported on populations initiating ART and, if available, searched conference archives using the key words of “initiation”, “naïve”, “re-initiate”, “newly” and “experience”. We have updated our methods section to reflect this.
The choice of study types to include (line 156) is good as is your use of citation searching in any systematic reviews found by your search. Those reviews could be cited or at least have the numbers indicated on the PRISMA flow chart/Fig.1	All systematic reviews identified in the search were checked for papers published ≥ 2018 and compared to our search results to confirm that we had already located them. Any that were missed in our original search were then added to the search results. Because a very large number of systematic reviews (>60) were checked, we have now listed them in a supplemental file (S4), rather than in the main reference section of the manuscript.
The screening methods described on 177 forward are solid and clearly explained.	Thank you.

Comment	Response
Line 184-5 states that the results of the search were documented in accordance with PRISMA-P. PRISMA-P is for reporting protocols. Did you use PRISMA-S? It recommends describing which version of databases you used you used (eg Embase) and which databases within multi-database platforms you used (eg Web of Science) which were not included.	With apologies, we did use PRISMA-S. As mentioned above, we have now included database versions as recommended.
The search date is not mentioned. Although it could be the same as the end date for the searches, March 31, 2022, this is not clear. That date is nearly a year ago. Considering the small amount of evidence you found, rerunning the searches to find results published since March 2022 is recommended. Since you used Rayyan, which will identify duplicates, reviewing only new results would be simple.	The initial search was conducted on 31 March, with an update for our initial submission occurring on September 15. We agree that due to time passed since submission, an update was warranted and performed this search on the 11th of July 2023, including all papers published till that date. We have edited lines 151-156 to clarify this.
I recommend having the search strategy rewritten by a librarian experienced with evidence synthesis/systematic review searching. Using Rayyan to identify duplicates, you can avoid rescreening. This should improve the recall of on-topic studies as well as finding any studies from the past year. You may end up with similar conclusions, but they will be based on better methods and evidence.	Thank you for this comment. Our original and updated search strategies were drafted and conducted by David Flynn, Assistant Director of Library and Information Management Education for Boston University Medical Center. You may review his credentials here: https://orcid.org/0000-0002-7494-2098. He has been added as a co-author to the manuscript.

Reviewer 2

Dr. Gert U Van Zyl, Stellenbosch University Faculty of Medicine and Health Sciences

Comments to the Author:

The topic of prior antiretroviral exposure in patients presenting for ART initiation is very topical and important. Prior drug exposure may influence treatment outcomes and may result in a much higher virologic failure rate or rate of drug resistance than in patients who are truly therapy naïve. This is true even for high genetic barrier regimens: Whereas the incidence of virologic failure and especially drug resistance is exceeding rare in therapy naïve patients receiving first line regimens that include dolutegravir (DTG) as anchor drug, it is not uncommon in patients with virologic failure on a DTG containing regimen, who have had prior virologic failure on different regimens. Prior ART exposure could therefore result in lower real-life effectiveness of particular regimens than the expected efficacy observed in clinical trials of treatment naïve participants.

The manuscript is overall well written. It followed a structured and clear search plan. The authors report heterogeneity in the methods to establish prior ART use. The proportion of prior ART use varied between 2% and 53%. In studies with lower proportions this may represent an underestimation due to relying on self-report. The authors also make an important comment, stating that reinitiating patients include patients who have experienced obstacles to care and may require services to improve their outcomes.

We thank the Reviewer for this overview.

Comment that could be considered for further improvement:

Comment	Response
Please add a brief comment on the potential impact of having a large proportion of non-naïve patients on apparent first-line ART treatment success.	We thank the reviewer for highlighting the importance of distinguishing potential difference in outcomes between naïve and non-naïve ART initiators. We have added a sentence about the potential impact on treatment outcomes.

Reviewer 3

Dr. Jienchi Dorward, University of Oxford, Centre for the Aids Programme of Research in South Africa

Comments to the Author:

Thank you for opportunity to review this systematic review of studies assessing the proportion of people who are ART naïve out of those initiating ART. This is a well written study of an important topic and appears to have been well conducted. I have few mainly minor comments.

We thank the Reviewer for these comments.

Comment	Response
Major	
Throughout the manuscript, It would help to be a bit clearer about the different measures used in the	Thank you. We agree that this is an important point and was not sufficiently addressed in the

studies, and the difference in what they are measuring ie currently or recently on ART (measured by suppressed VL and/or measured ART metabolites) versus those who at some point have been on ART (which could have been a long time in the past and so no ART metabolites would be detected). I realise that the lack of studies using the different methods means that analysis/stratification based on these different measures is not possible, but it would help with moving the field forwards if this difference was clearly set out in the introduction, and discussed in the discussion.	original manuscript. We have added a note about the time frame associated with viral suppression versus other measures of prior exposure to the discussion session.
Minor	
Methods: Why were only studies after 2018 included? If Universal treatment started in 2016, then it is feasible that useful data could have been reported in 2017	Universal treatment was adopted over the course of 2016, with some countries only issuing guidelines allowing universal eligibility towards the end of that year. South Africa, for instance, began implementation in September 2016 (https://sahivsoc.org/Files/22%208%2016%20Circular%20UTT%20%20%20Decongestion%20CCMT%20Directorate.pdf). For this reason, starting with publications in 2018 seemed like a reasonable way to capture the universal treatment era.
Commendable effort with full text review of >300 manuscripts! I am also surprised that there are so few relevant studies. I'm not a search expert but would the search terms have captured studies that did not mention sub-Saharan Africa, but only mentioned a country by name?	We thank the reviewer for highlighting this. We have updated our search to include country names.
Line 107 "By definition, treatment re-initiators previously faced barriers to retention in care that they were unable to overcome." – what about people who relocate – its not necessarily a barrier that they have to overcome?	This is a good point. We agree that unknown transfers is a limitation for nearly all observational data in this setting. The fact that these patients, however, present for initiation rather than continuation of services suggests that, at the very least, they are transferring informally ("silent transfers") and not remaining engaged in care as we define it. It is likely correct, though, that for a subset of those who relocate, treatment is not interrupted, or only for a very short time. For clarify, we have revised this sentence to read "By definition, most treatment re-initiators previously faced barriers to retention in care that they were unable to overcome in time to retain continuity of treatment." We have also added the possibility of

	silent transfer without treatment interruption to our list of study limitations.
For those with suppressed viral loads, could there have been	We are not certain what the Reviewer is referring to here.
Table 2: "Sample included 16-17 year olds" is it possible to give the proportion who were this age in the sample, which will help reader to determine how important this is. Likewise for other studies which included children	In the study, the proportion of 16-17 year olds is not reported. "Adults" in this study included anyone 16 or older. We therefore allowed this definition and included the study in the review, rather than excluding it because our own definition of adult is 18 or older. The same pertained to Genet et al, which included participants age 12 and older, but in this case the paper reported that those under 18 comprised only 7% of the sample. We have clarified these points in footnotes to the tables.
Studies excluded for the wrong population: "Population for main analysis excluded those with ART experience or required a minimum follow-up period after treatment initiation." But it seems like some studies which excluded people who weren't ART naïve, or required minimum follow-up e.g. Dorward were included. A bit more explanation about why some were excluded and some included would help.	Thank you for catching this discrepancy. While we originally had planned to exclude studies with a required minimum follow up period, we realized over the course of the review that this would omit potentially relevant sources and that the limitations introduced by the required follow up period could simply be noted in our manuscript. We therefore did include studies with a required minimum follow up in the review but neglected to make the change in the methods section. We have now revised supplemental table 3 to accurately reflect review procedures.

VERSION 2 – REVIEW

REVIEWER	Reynolds, John University of Miami Miller School of Medicine, Calder Memorial Library
REVIEW RETURNED	30-Aug-2023

GENERAL COMMENTS	Thank you for the opportunity to review your evidence synthesis project. It seems like a useful project and the search methods and reporting have been greatly improved and clarified since the first submission. I have a few minor comments: The description of the review type still says "rapid" on line 132 while the rest of the paper has been corrected to "systematic." In your comment you note that David Flynn, MSLIS has been added as a coauthor, but I don't see his name listed on the manuscript. On line 135 you have removed WHO rapid review guidelines and added PRISMA guidelines. PRISMA is a reporting guideline
--

	(hence the name Preferred Reporting Items for Systematic Reviews and Meta-analyses) and is insufficient for conducting a review. An appropriate guideline for conduct, as opposed to reporting, would be the Methodological Expectations of Cochrane Intervention Reviews (MECIR) https://community.cochrane.org/mecir-manual and its use could be phrased like "The systematic review was conducted with guidance from the Methodological Expectations of Cochrane Intervention Reviews Manual and reported in accordance with the Preferred Reporting Items for Systematic Reviews and Meta-analyses." It looks to me like you followed much of the recommendations there. Otherwise, simply say that your review is reported according to PRISMA rather than conducted. At line 165 you indicate that searches were done through March 31, 2023, but you have updated the search since then, as noted on line 169. It might be clearer to simply use the latter date. Similarly, it might not be necessary to document that you tried using Google Scholar without good results, and simply state that you searched the conference websites directly. It would also be helpful to add your conference searching methods to the supplemental table, including the names of the conferences and URLs, since those were part of your data acquisition process also. Although it is an excellent resource, it is probably not necessary to cite the ISSG Search Filter Resource (reference 10) since it is a listing of collected resources, not the creator of the filter you used. Also, the "Afrique subsaharienne" filter you used is very good, but slightly out of date – Swaziland changed its name to Eswatini in 2018. I seriously doubt that this will impact your findings, but you should check by adding Eswatini terms to the filter and using the NOT operator to compare the results, or simply running a search with Eswatini as a subject and title/abstract/keyword term combined with your infection, treatment and compliance terms and screening the results. If you plan to do any future evidence synthesis research, consider having a librarian take advantage of the PRESS process, which would have caught many of the corrections in your search prior to submitting your manuscript. See McGowan J, Sampson M, Salzwedel DM, Cogo E, Foerster V, Lefebvre C. PRESS Peer Review of Electronic Search Strategies: 2015 Guideline Statement. J Clin Epidemiol. 2016 Jul;75:40-6. doi: 10.1016/j.jclinepi.2016.01.021. Epub 2016 Mar 19. PMID: 27005575.
--	---

REVIEWER	Dorward, Jienchi University of Oxford, Nuffield Department of Primary Care Health Sciences
REVIEW RETURNED	10-Aug-2023

GENERAL COMMENTS	It would have been helpful if the authors could have provided the actual text of the changes that they made in thier response, or at least the line numbers, to save the reviewer from having to look up the changes themselves. But overall my comments have been addressed adequately.
--

VERSION 2 – AUTHOR RESPONSE

Reviewer: 1

Prof. John Reynolds, University of Miami Miller School of Medicine

Comment	Response
Thank you for the opportunity to review your evidence synthesis project. It seems like a useful project and the search methods and reporting have been greatly improved and clarified since the first submission.	Thank you.
The description of the review type still says "rapid" on line 132 while the rest of the paper has been corrected to "systematic."	Thank you for highlighting this oversight. We've corrected line 132 to read systematic review.
In your comment you note that David Flynn, MSLIS has been added as a coauthor, but I don't see his name listed on the manuscript.	We thank the reviewer for bringing this to our attention. The author with his affiliation has been added.
On line 135 you have removed WHO rapid review guidelines and added PRISMA guidelines. PRISMA is a reporting guideline (hence the name Preferred Reporting Items for Systematic Reviews and Meta-analyses) and is insufficient for conducting a review. An appropriate guideline for conduct, as opposed to reporting, would be the Methodological Expectations of Cochrane Intervention Reviews (MECIR) https://community.cochrane.org/mecir-manual and its use could be phrased like "The systematic review was conducted with guidance from the Methodological Expectations of Cochrane Intervention Reviews Manual and reported in accordance with the Preferred Reporting Items for Systematic Reviews and Meta-analyses." It looks to me like you followed much of the recommendations there. Otherwise, simply say that your review is reported according to PRISMA rather than conducted.	We thank the reviewer for this suggestion. We have edited our text to indicate that we conducted the review with guidance from the MECIR and that our results are reported using PRISMA (lines 147-149).
At line 165 you indicate that searches were done through March 31, 2023, but you have updated the search since then, as noted on line 169. It might be clearer to simply use the latter date. Similarly, it might not be necessary to document that you tried using Google Scholar without good results, and simply state that you searched the conference websites directly. It would also be helpful to add your conference searching methods to the supplemental table, including the names of	Thank you. We have edited this section (lines 168-178) for clarity. We've updated the supplemental material with our search strategy of the conferences we searched.

Comment	Response
the conferences and URLs, since those were part of your data acquisition process also.	
Although it is an excellent resource, it is probably not necessary to cite the ISSG Search Filter Resource (reference 10) since it is a listing of collected resources, not the creator of the filter you used.	Thank you, we've removed this citation.
Also, the "Afrique subsaharienne" filter you used is very good, but slightly out of date – Swaziland changed its name to Eswatini in 2018. I seriously doubt that this will impact your findings, but you should check by adding Eswatini terms to the filter and using the NOT operator to compare the results, or simply running a search with Eswatini as a subject and title/abstract/keyword term combined with your infection, treatment and compliance terms and screening the results.	Thank you, we acknowledge that using established hedges may have its limitations, as you mentioned. Our search using the old name identified 4 publications from Eswatini and including Eswatini as a term in our search did not yield any additional articles.
If you plan to do any future evidence synthesis research, consider having a librarian take advantage of the PRESS process, which would have caught many of the corrections in your search prior to submitting your manuscript. See McGowan J, Sampson M, Salzwedel DM, Cogo E, Foerster V, Lefebvre C. PRESS Peer Review of Electronic Search Strategies: 2015 Guideline Statement. J Clin Epidemiol. 2016 Jul;75:40-6. doi: 10.1016/j.jclinepi.2016.01.021. Epub 2016 Mar 19. PMID: 27005575.	We thank the reviewer for making us aware of this resource.